# Modeling the relative risk of incidence and mortality of select vaccine-preventable diseases by wealth group and geographic region in Ethiopia

**Sarah Bolongaita**[1], **Dominick Villano**[1], **Solomon Tessema Memirie**[2], **Mizan Kiros Mirutse**[3], **Alemnesh H. Mirkuzie**[4], **Sophia Comas**[1], **Eva Rumpler**[5], **Stephanie M. Wu**[6], **Ryoko Sato**[1], **Angela Y. Chang**[7], **Stéphane Verguet**[1] *

1 Department of Global Health and Population, Harvard T.H. Chan School of Public Health, Boston, Massachusetts, United States of America, 2 Addis Center for Ethics and Priority Setting, College of Health Sciences, Addis Ababa University, Addis Ababa, Ethiopia, 3 Ministry of Health, Federal Democratic Republic of Ethiopia, Addis Ababa, Ethiopia, 4 National Data Management Center for Health,Ethiopian Public Health Institute, Addis Ababa, Ethiopia, 5 Department of Epidemiology, Harvard T.H. Chan School of Public Health, Boston, Massachusetts, United States of America, 6 Department of Biostatistics, Harvard T.H. Chan School of Public Health, Boston, Massachusetts, United States of America, 7 Danish Institute for Advanced Study, University of Southern Denmark, Odense, Denmark

* verguet@hsph.harvard.edu

**Data Availability Statement:** Data are available from the Demographic Health Surveys (https://dhsprogram.com).

## Abstract

Immunization is one of the most effective public health interventions, saving millions of lives every year. Ethiopia has seen gradual improvements in immunization coverage and access to child health care services; however, inequalities in child mortality across wealth quintiles and regions remain persistent. We model the relative distributional incidence and mortality of four vaccine-preventable diseases (VPDs) (rotavirus diarrhea, human papillomavirus, measles, and pneumonia) by wealth quintile and geographic region in Ethiopia. Our approach significantly extends an earlier methodology, which utilizes the population attributable fraction and differences in the prevalence of risk and prognostic factors by population subgroup to estimate the relative distribution of VPD incidence and mortality. We use a linear system of equations to estimate the joint distribution of risk and prognostic factors in population subgroups, treating each possible combination of risk or prognostic factors as computationally distinct, thereby allowing us to account for individuals with multiple risk factors. Across all modeling scenarios, our analysis found that the poor and those living in rural and primarily pastoralist or agrarian regions have a greater risk than the rich and those living in urban regions of becoming infected with or dying from a VPD. While in absolute terms all population subgroups benefit from health interventions (e.g., vaccination and treatment), current unequal levels and pro-rich gradients of vaccination and treatment-seeking patterns should be redressed so to significantly improve health equity across wealth quintiles and geographic regions in Ethiopia.

**Funding:** SV - We acknowledge funding support from Gavi, the Vaccine Alliance (https://www.gavi.org/). The funders had no role in study design, data collection and analysis, decision to publish, or preparation of the manuscript.

**Competing interests:** The authors have declared that no competing interests exist.

## Introduction

Immunization is one of the most effective public health interventions, saving millions of lives every year [1,2]. Globally and in Ethiopia, access and use of vaccines has progressively expanded, and several new vaccines have been introduced, including those against major killers such as pneumococcal pneumonia, rotavirus diarrhea, cervical cancer, and meningitis [3]. Global coverage of the first dose of measles vaccine among one-year-olds, for example, has increased from 72% in 2000 to 85% in 2019 [4,5]. Similarly, in Ethiopia, measles vaccination coverage improved from 26% to 52% over the same time period [6,7]. Yet, despite impressive progress over the past two decades, immunization coverage gaps persist between and within countries, across geographic locations and socioeconomic status [3,8]. Improving immunization coverage and equity were among the strategic objectives of the Global Vaccine Action Plan (GVAP) and remain strategic priorities in the Immunization Agenda 2030 (IA2030) [3].

Underserved populations often carry a heavier burden of vaccine-preventable diseases (VPD) and lack access to basic health care. Despite gradual improvements in coverage of child health care services in Ethiopia, inequality in child mortality and access to care between urban and rural dwellers and across wealth quintiles remain persistent (Fig A in S1 Text) [9]. Across the rural-urban divide, there were 64 under-five deaths per 1,000 live births in rural areas compared to 46 in urban areas, and those in the poorest quintile had an under-five mortality rate of 77 (per 1,000 live births) compared to 46 among those in the wealthiest quintile in 2019 [7].

These outcomes relate to differential access to vaccines, as well as health-seeking behavior. In 2019, only 26% of children aged 12–23 months in the poorest quintile received all eight basic immunizations (Bacille Calmette-Guérin vaccine; three doses of DPT-containing vaccine against diphtheria, pertussis, tetanus; three doses of polio vaccine; and one dose of measles-containing vaccine), compared to 67% among the wealthiest quintile. In 2016, the poorest households sought care from a healthcare provider 40% of the time for symptoms of childhood diarrhea and 25% of the time for symptoms of childhood pneumonia, whereas the richest households sought care 61% and 44% of the time, respectively [9]. Lower service utilization among the poorest occurs despite an increased risk of diarrhea and pneumonia among children from this wealth group [9].

Vaccines may have large distributional impacts that could improve health equity [10]. Chang and colleagues analyzed the potential equity impact of vaccines on averting mortality in 41 low- and middle-income countries (LMICs) and found that benefits across ten vaccine antigens could accrue predominantly in the lowest income quintiles [10]. To estimate the distribution of vaccine-averted incidence and mortality for each antigen across socioeconomic strata, the authors quantified the contribution of sets of risk and prognostic factors (and their prevalence differences by quintile) that can be used to estimate the likelihood of contracting and dying from each disease. Risk factors are behaviors or traits associated with causing disease, whereas prognostic factors are behaviors or traits that influence the outcome of the disease once infected. Quintile-specific vaccination coverage rates and quintile-specific treatment utilization rates were then applied to determine the number of disease-specific cases and deaths averted, respectively.

Ethiopia is a low-income country with the second largest population in Africa [11]. The country's Expanded Programme on Immunization (EPI) is a core priority of the national Health Sector Transformation Plan (HSTP), wherein vaccines are provided routinely in health facilities, as well as in outreach and mobile health facilities and through vaccination campaigns [12]. Ethiopia has developed several different strategies to improve immunization coverage including utilization of the women (or health) development army (WDA), a women-centered community movement designed to improve the implementation capacity of the health system in rural areas (especially agrarian societies) through community engagement, and

comprehensive multi-year immunization plans (cMYP), which encompass all components of immunization services (e.g., service delivery, vaccine supply, quality and logistics, disease surveillance, advocacy, social mobilization and communication, and program management) [12,13]. However, effectively translating these strategies and plans into action has been challenging and community engagement has declined in recent years, with waning capacity and acceptability among WDA leaders and low acceptance by community members [12,13].

Since Ethiopia's introduction of the EPI in 1980, immunization coverage has progressively increased; however, there remains a substantial coverage gap and inequality in coverage by socioeconomic stratum and by geographic location [6,7,9,14,15]. Nationally, the proportion of children who were fully immunized was 41% in 2019, but this varies drastically [7]. Only 20% of children in the Afar region (a predominately pastoralist population) are fully vaccinated, compared to 83% in Addis Ababa (an entirely urban population) [7]. Similarly, disparities in immunization coverage are also observed by wealth quintile [7].

Chang and colleagues [10,16] have proposed a first modeling approach to assess the distributional impact of vaccines on VPD incidence and mortality; however, their methodology did not yet account for the joint distribution of risk factors (i.e., the overlap between risk factors, that is accounting for individuals with multiple risk factors) and did not include variation by geographic region, neither did it examine the country of Ethiopia specifically. In this paper, we build on this preliminary work and significantly augment it in assessing the relative distribution of VPD incidence and mortality across socioeconomic strata and by geographic region in Ethiopia, taking into account the joint distribution of risk factors for these diseases.

## Methods

We examined four major VPDs in Ethiopia: rotavirus diarrhea, human papillomavirus (HPV), measles, and pneumonia (pneumococcal and *Haemophilus influenzae* type b, Hib), as well as their five corresponding vaccines: rotavirus vaccine (RV, against rotavirus diarrhea), HPV vaccine, measles-containing vaccine (MCV), pneumococcal conjugate vaccine (PCV, against pneumococcal pneumonia), and penta-3 vaccine (against Hib pneumonia). We evaluated all vaccines in Ethiopia's EPI for inclusion in the analysis; however, only rotavirus diarrhea, measles, pneumonia, and HPV were selected as VPDs for inclusion due to data availability. Data on risk and prognostic factors for these diseases (with the exception of HPV) is available from the Demographic and Health Survey (DHS) disaggregated by wealth quintile and region [9].

### Risk and prognostic factors

For each VPD, we selected a number of risk and prognostic factors. Diarrhea, measles, and pneumonia have well-studied and established risk and prognostic factors, but there is more uncertainty surrounding these factors for HPV [17–19]. A thorough literature review was conducted to gather evidence on the risk and prognostic factors for HPV, with factors evaluated for use in the analysis based on the strength of evidence and the availability of data (disaggregated by wealth quintile and region) in the Ethiopian DHS [9]. Prognostic factors for HPV-related disease outcomes, notably cervical cancer, are unavailable in disaggregate and we therefore did not analyze HPV-related deaths. Sections 2 and 3 in S1 Text provide detailed information on our methods and summarize the risk and prognostic factors selected for each VPD (see Table 1 for a brief summary).

### Distributional modeling

Our approach substantially extends a previous methodology, which utilizes the population attributable fraction (PAF) and differences in the prevalence of risk and prognostic factors

**Table 1. Selected risk and prognostic factors for diarrhea, measles, pneumonia, and HPV.** See Section 3 of S1 Text for more detailed methodology on risk factor selection.

| Disease | Risk factors | Prognostic factors |
|---|---|---|
| Diarrhea* | Stunting [20–23]<br>Underweight [20–22,24]<br>Unsafe sanitation [21,22,24]<br>Wasting [20–22,24] | Stunting [20–22,24]<br>Underweight [20–22,24]<br>Wasting [20–22,24] |
| Measles* | Stunting [20,21]<br>Underweight [20,21]<br>Vitamin A deficiency [20,21,25]<br>Wasting [20,21] | Stunting [20,21]<br>Underweight [20,21]<br>Vitamin A deficiency [20,21,25]<br>Wasting [20,21] |
| Pneumonia* | Underweight [20,21,26]<br>Vitamin A deficiency [20,21,23,26]<br>Wasting [20,21,26] | Stunting [20,21,26]<br>Underweight [20,21,23]<br>Vitamin A deficiency [20,21,23,26]<br>Wasting [20,21,26] |
| HPV | Age at first sexual intercourse [27]<br>Number of lifetime sexual partners [27–29] | Not analyzed** |

*Risk and prognostic factors for measles, diarrhea, and pneumonia were adapted from Chang et al., 2018 [16].

**Prognostic factors for HPV disease outcomes, notably cervical cancer, are unavailable in disaggregate and we therefore did not analyze HPV-related deaths.

across wealth groups to estimate the distribution of VPD incidence and mortality across those wealth quintiles [16]. In this paper, we augment this methodology by using a linear system of equations to estimate the joint distribution of risk and prognostic factors in population sub-groups (i.e. wealth quintile and geographic region). To compute the joint distribution of risk and prognostic factors, we treat each possible combination of risk or prognostic factors as computationally distinct, thereby allowing us to account for individuals with multiple risk factors.

For a given VPD, a simulated population (derived from DHS data) is divided according to all the possible risk factor combinations. The incidence and mortality of the VPD are estimated for each risk factor combination by minimizing a constrained linear system. The parameters of the linear systems are functions of the relative risks, the proportion of the risk factor combi-nations within the population, and the proportion of cases or deaths attributable to the disease. Optimization of VPD incidence for each risk factor combination was performed starting from a random admissible probability corresponding to either cases or attributable deaths. From this random starting point, we identified a nearby probability combination that provided a bet-ter fit to the parameters of the system. Then, from this second point, an improved nearby fit was identified, providing a third point that provided a better fit than the first two points. The process was repeated until a minimizer was identified (further details are provided in Section 2 of S1 Text).

After optimizing the multidimensional risk factor incidences, each individual in the simu-lated population was assigned a probability of morbidity (case) or mortality (death) according to their risk factor profile, giving rise to a probability distribution that can be further analyzed by wealth quintile, geographic region, or in totality. In order to estimate the impact of vaccina-tion across quintiles and regions, the expected number of cases predicted by the probability distribution was reduced according to vaccination coverage rates and efficacies (Table 2).

To estimate the distribution of mortality across quintiles and regions we employed three computational strategies. First, we applied estimates of quintile- and region-specific treat-ment-seeking behavior (e.g., care-seeking for diarrhea and lower respiratory infections, as reported in the DHS) by reducing the number of cases proportionally to estimated rates of

**Table 2. Vaccine and treatment efficacies for rotavirus diarrhea, HPV, measles, and pneumonia (pneumococcal and Hib) used in the model.**

| Disease | Vaccine | Vaccine efficacy | Treatment efficacy |
|---|---|---|---|
| Rotavirus diarrhea | Rotavirus vaccine | 0.50 (0.11–0.72) [30] | 0.93 (0.83–0.98) [31] |
| Measles | Measles-containing vaccine | 0.85 (0.83–0.87) [32] | 0.62 (0.52–0.82) [32] |
| Pneumonia | | | 0.70 (0.52–0.82) [33] |
| Pneumococcal (33% of pneumonia [30]) | Pneumococcal conjugate vaccine | 0.58 (0.29–0.75) [33] | |
| Hib (22% of pneumonia [30]) | Penta-3 vaccine | 0.93 (0.83–0.97) [34] | |
| HPV | HPV vaccine | 1.00 | Not analyzed* |

* Prognostic factors for HPV disease outcomes, notably cervical cancer, are unavailable in disaggregate and we therefore did not analyze HPV-related deaths. As such, we did not utilize a treatment efficacy parameter for HPV in the model.

treatment-seeking multiplied by the efficacy of treatment (as provided by the literature, Table 2). Next, we adjusted the proportion of cases using quintile- and region-specific estimates of under-five mortality rates. Finally, the proportion of mortality was computed entirely independently from cases (i.e., using prognostic factors), with the impact of vaccination and treatment-seeking behavior implemented analogously. The results presented here report the average results of the second two strategies, which incorporate differences in unvaccinated incidence and mortality gradients. Full results are provided in Section 4 of S1 Text.

The model assigns a risk of incidence and mortality due to the various VPDs by wealth quintile and geographic region. The incidence and mortality risk gradients are normalized such that the quintile or region with the lowest risk is assigned a value of one. A value of two, for example, would therefore indicate that individuals in that given population subgroup (quintile or region) are twice as likely to die from the VPD than individuals from the lowest risk population subgroup (quintile or region, respectively), with a value of one. Results are reported for three scenarios: a "counterfactual" scenario (no vaccination or treatment-seeking), a baseline scenario (current vaccination coverage and treatment-seeking behavior), and a maximized vaccination scenario in which all population subgroups were set to have the same vaccination coverage as the population subgroup with the highest vaccination coverage.

## Results

The proportional risk distribution of VPD incidence and mortality across wealth quintiles and geographic regions was estimated for Ethiopia for: a "counterfactual" scenario, a baseline scenario, and a maximized vaccination scenario. Across all VPDs and scenarios, the risk gradients across wealth quintiles show an inverse relationship with wealth, such that the poor have a greater risk of becoming infected with or dying from a VPD (Fig 1). Likewise, by geographic region, regions with more rural and primarily pastoralist and agrarian populations have a greater risk of VPD than regions with richer and predominantly urban populations (Figs 2 and 3).

In the "counterfactual" scenario, VPD incidence and mortality is estimated based on the distribution of risk and prognostic factors and does not include health interventions (i.e., vaccination coverage and treatment-seeking behavior). Compared to the richest quintile, the poorest quintile has 1.78 (95% uncertainty range (UR): 1.65–1.93) times greater risk of rotavirus diarrhea incidence, 1.83 (1.52–2.23) times greater risk of measles incidence, 2.07 (1.97–2.16) times greater risk of pneumococcal and Hib pneumonia incidence, and 1.13 (1.09–1.17) times greater risk of HPV incidence (Table 3). Compared to the gradients observed by wealth quintile, the relative risk gradients by region are much larger. For example, Afar, which has a

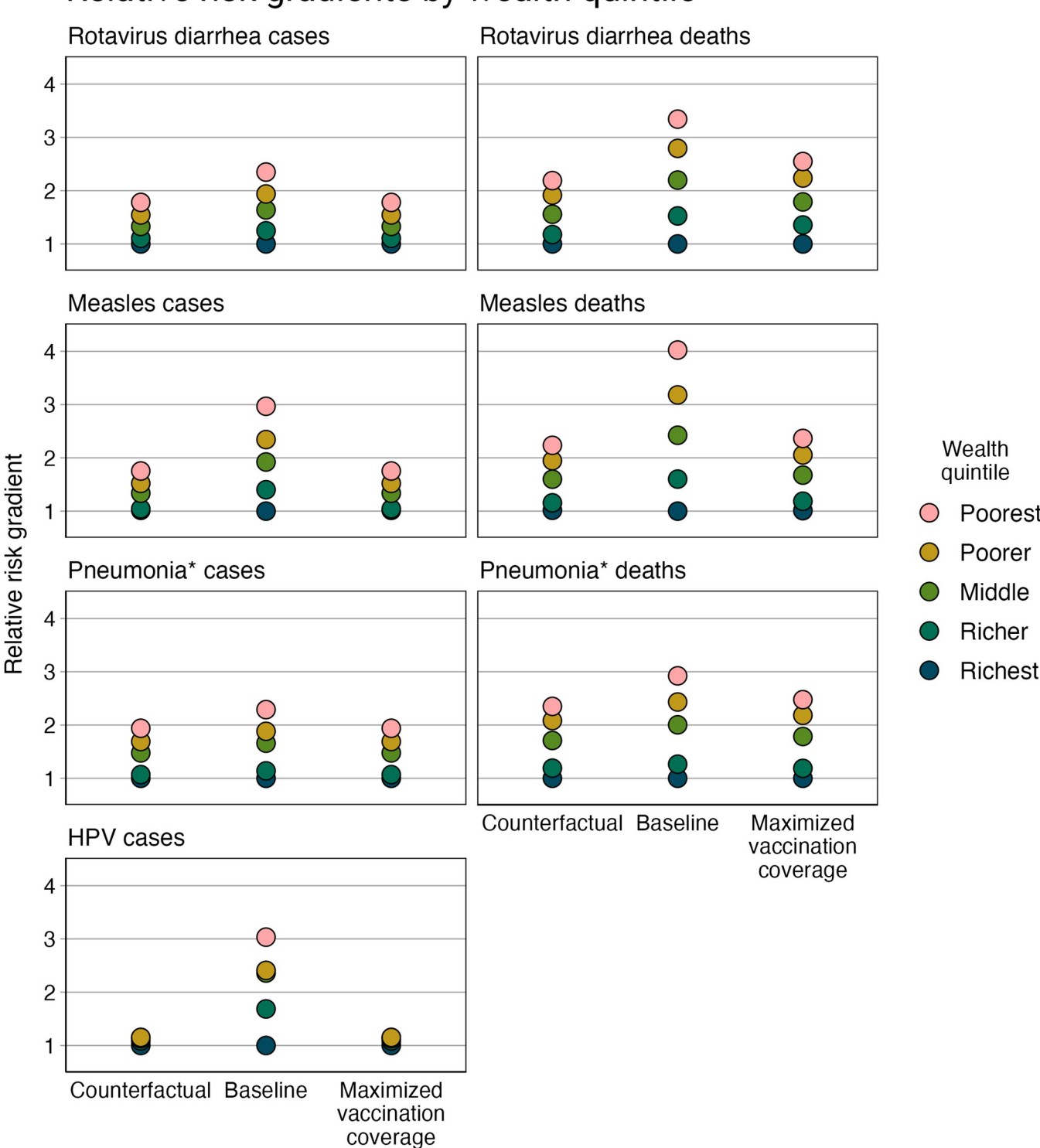

\* Pneumococcal and Hib

**Fig 1. Relative risk gradients of rotavirus diarrhea, measles, pneumonia (pneumococcal and Hib), and HPV incidence and mortality by wealth quintile.**
The models for all three scenarios include the distribution of risk and prognostic factors. The "counterfactual" scenario presents a scenario without vaccination or treatment-seeking behavior; the baseline scenario presents a scenario with current vaccination coverage and treatment-seeking behavior; and the maximized

vaccination coverage scenario presents a scenario in which vaccination coverage in all population subgroups is increased to the level of vaccination coverage of the subgroup with the greatest vaccination coverage (e.g., all quintiles are modeled as having the same level of vaccination coverage as the richest quintile).

predominantly rural and pastoralist population, has 3.79 (3.42–4.28) times greater risk of rotavirus diarrhea incidence, 4.50 (2.67–7.60) times greater risk of measles incidence, 5.99 (5.12–6.96) times greater risk of pneumococcal and Hib pneumonia incidence, and 1.48 (1.33–1.68) times greater risk of HPV incidence compared to Addis Ababa (the richest region with an entirely urban population). And the relative risk gradients, by both wealth quintile and region, are steeper when evaluating disease mortality.

The relative VPD incidence and mortality gradients, in the baseline scenario in which vaccination coverage and treatment-seeking behavior are now included, indicate greater relative inequalities. Applying quintile- and region-specific vaccination coverage rates results in steeper relative incidence gradients (compared to the counterfactual scenario). This impact is compounded when examining relative mortality gradients, which also take into account quintile- and region-specific treatment-seeking behavior. Compared to the richest quintile, the poor have 3.34 (2.75–4.26) times the risk of mortality from rotavirus diarrhea, 4.23 (3.45–5.23)

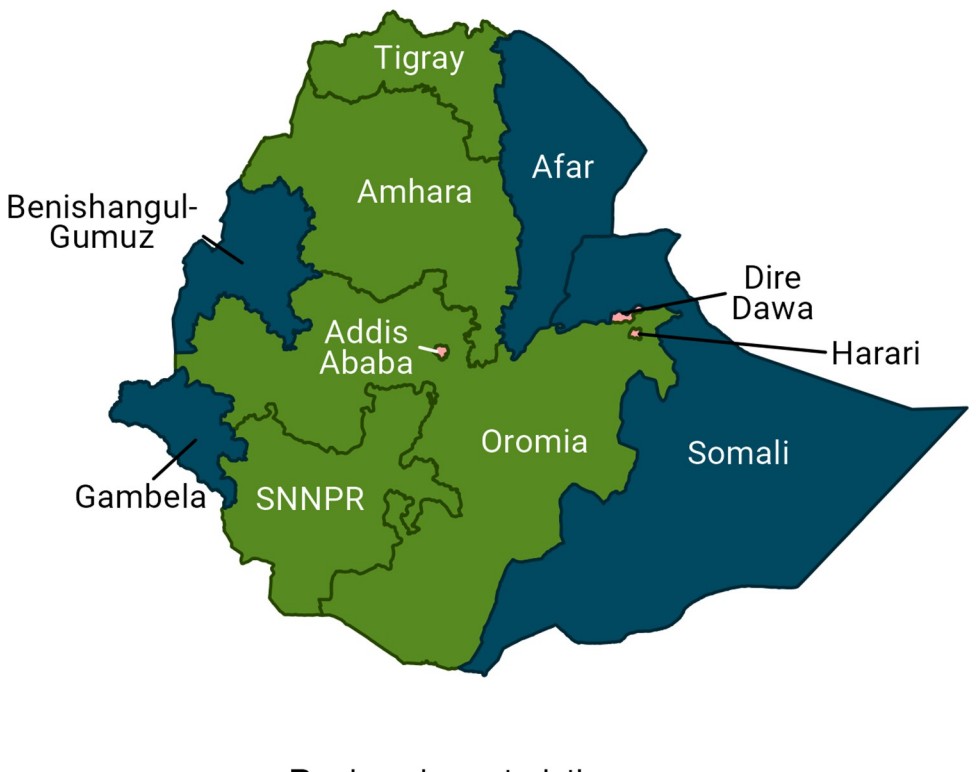

**Fig 2. Selected sociodemographic characteristics of Ethiopian regions.** Base map data is available from GADM: https://gadm.org/download_country.html.

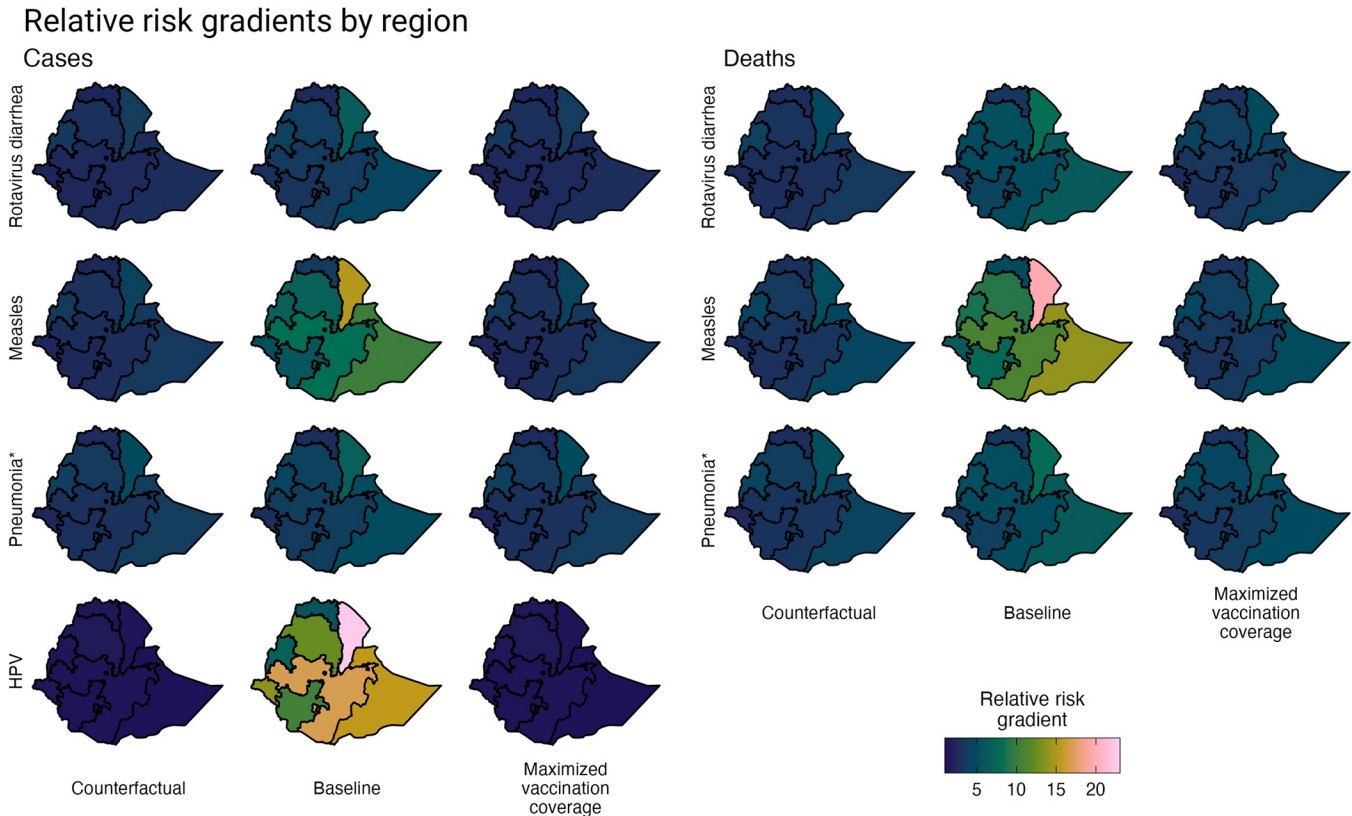

**Fig 3. Relative risk gradients of rotavirus diarrhea, measles, pneumonia (pneumococcal and Hib), and HPV incidence and mortality across three scenarios by geographic region.** The three scenarios include the distribution of risk and prognostic factors. The "counterfactual" scenario presents a scenario without vaccination or treatment-seeking behavior; the baseline scenario presents a scenario with current vaccination coverage and treatment-seeking behavior, and the maximized vaccination coverage scenario presents a scenario in which vaccination coverage in all population subgroups is increased to the level of vaccination coverage of the subgroup with the greatest vaccination coverage (e.g., all quintiles are modeled as having the same level of vaccination coverage as the richest quintile). Base map data is available from GADM: https://gadm.org/download_country.html.

times the risk of mortality from measles, and 3.05 (2.84–3.32) times the risk of mortality from pneumococcal and Hib pneumonia. Regionally, compared to Addis Ababa, the population in Afar has 8.36 (5.83–13.38) times the risk of mortality from rotavirus diarrhea and 9.27 (7.87–11.07) times the risk of mortality from pneumococcal and Hib pneumonia.

In the maximized vaccination scenario, the relative risk gradients for VPD incidence return to the relative gradients observed in the counterfactual scenario, which demonstrates that increasing and equalizing vaccination coverage in Ethiopia would significantly improve health equity. As all quintiles and regions have the same vaccination coverage under this scenario, only the underlying inequalities in the distribution of risk and prognostic factors therefore contribute to the relative incidence gradients. Differences remain, however, for the relative mortality gradients, as treatment-seeking behavior was not modified in this scenario (i.e., treatment seeking was maintained similar across the baseline and maximized vaccination coverage scenarios).

## Discussion

In this research we build and expand upon previous methods, which utilize the PAF and differences in the prevalence of risk and prognostic factors across population subgroups to estimate

**Table 3. Estimated relative risk distribution of (A) rotavirus diarrhea incidence and mortality, (B) measles incidence and mortality, (C) pneumonia (pneumococcal and Hib) incidence and mortality, and (D) HPV incidence across three scenarios by wealth quintile and geographic region.** The three scenarios include the distribution of risk and prognostic factors. The "counterfactual" scenario presents a scenario without vaccination or treatment-seeking behavior, the baseline scenario presents a scenario with current vaccination coverage and treatment-seeking behavior, and the maximized vaccination coverage scenario presents a scenario in which vaccination coverage in all population subgroups is increased to the level of vaccination coverage of the subgroup with the greatest vaccination coverage (e.g., all quintiles are modeled as having the same level of vaccination coverage as the richest quintile).

| (A) Rotavirus diarrhea | | | | | | |
|---|---|---|---|---|---|---|
| | "Counterfactual" | | Baseline | | Maximized vaccination coverage | |
| | Incidence | Mortality | Incidence | Mortality | Incidence | Mortality |
| **Quintile** | | | | | | |
| Poorest | 1.78 (1.65–1.93) | 2.19 (2.04–2.36) | 2.34 (1.91–3.01) | 3.34 (2.75–4.26) | 1.78 (1.65–1.93) | 2.55 (2.34–2.76) |
| Poorer | 1.55 (1.46–1.66) | 1.92 (1.82–2.05) | 1.93 (1.64–2.41) | 2.79 (2.38–3.44) | 1.55 (1.46–1.66) | 2.24 (2.09–2.41) |
| Middle | 1.33 (1.29–1.38) | 1.56 (1.52–1.62) | 1.64 (1.42–2.00) | 2.19 (1.91–2.66) | 1.33 (1.29–1.38) | 1.79 (1.72–1.87) |
| Richer | 1.11 (1.07–1.14) | 1.18 (1.15–1.21) | 1.24 (1.14–1.41) | 1.52 (1.39–1.72) | 1.11 (1.07–1.14) | 1.36 (1.30–1.41) |
| Richest | 1.00 (1.00–1.00) | 1.00 (1.00–1.01)* | 1.00 (1.00–1.00) | 1.00 (1.00–1.00) | 1.00 (1.00–1.00) | 1.00 (1.00–1.00) |
| **Region** | | | | | | |
| Addis Ababa | 1.00 (1.00–1.00) | 1.00 (1.00–1.00) | 1.00 (1.00–1.00) | 1.00 (1.00–1.00) | 1.00 (1.00–1.00) | 1.00 (1.00–1.00) |
| Afar | 3.79 (3.42–4.28) | 4.52 (4.09–5.05) | 6.54 (4.50–10.51) | 8.36 (5.83–13.38) | 3.79 (3.42–4.28) | 4.91 (4.43–5.48) |
| Amhara | 2.89 (2.62–3.25) | 3.30 (2.99–3.68) | 3.90 (3.07–5.47) | 5.32 (4.20–7.34) | 2.89 (2.62–3.25) | 3.98 (3.57–4.45) |
| Benishangul-Gumuz | 3.58 (3.17–4.13) | 4.14 (3.70–4.74) | 4.15 (3.48–5.21) | 4.84 (4.10–5.94) | 3.58 (3.17–4.13) | 4.19 (3.74–4.79) |
| Dire Dawa | 2.71 (2.53–2.94) | 2.97 (2.78–3.19) | 2.90 (2.65–3.26) | 2.98 (2.73–3.32) | 2.71 (2.53–2.94) | 2.79 (2.61–3.02) |
| Gambela | 2.21 (2.07–2.37) | 2.51 (2.34–2.68) | 2.94 (2.39–4.01) | 3.44 (2.79–4.67) | 2.21 (2.07–2.37) | 2.60 (2.42–2.78) |
| Harari | 2.41 (2.22–2.65) | 2.56 (2.38–2.78) | 3.18 (2.57–4.35) | 3.63 (2.96–4.90) | 2.41 (2.22–2.65) | 2.76 (2.56–3.00) |
| Oromia | 2.54 (2.34–2.78) | 2.91 (2.68–3.17) | 3.66 (2.81–5.28) | 4.94 (3.81–7.08) | 2.54 (2.34–2.78) | 3.46 (3.16–3.79) |
| SNNPR | 2.35 (2.15–2.64) | 2.69 (2.47–3.00) | 3.27 (2.54–4.73) | 4.22 (3.30–5.94) | 2.35 (2.15–2.64) | 3.05 (2.78–3.40) |
| Somali | 2.87 (2.68–3.12) | 3.61 (3.34–3.91) | 4.41 (3.28–6.70) | 6.40 (4.78–9.56) | 2.87 (2.68–3.12) | 4.21 (3.86–4.59) |
| Tigray | 2.33 (2.19–2.51) | 2.74 (2.59–2.91) | 2.63 (2.35–3.10) | 3.41 (3.06–3.99) | 2.33 (2.19–2.51) | 3.04 (2.86–3.25) |
| **(B) Measles** | | | | | | |
| | "Counterfactual" | | Baseline | | Maximized vaccination coverage | |
| | Incidence | Mortality | Incidence | Mortality | Incidence | Mortality |
| **Quintile** | | | | | | |
| Poorest | 1.83 (1.52–2.23) | 2.34 (1.94–2.85) | 3.09 (2.54–3.79) | 4.23 (3.45–5.23) | 1.83 (1.52–2.23) | 2.48 (2.03–3.05) |
| Poorer | 1.59 (1.36–1.91) | 2.05 (1.74–2.48) | 2.44 (2.07–2.98) | 3.37 (2.81–4.11) | 1.59 (1.36–1.91) | 2.15 (1.82–2.64) |
| Middle | 1.42 (1.26–1.69) | 1.70 (1.51–2.02) | 2.03 (1.79–2.45) | 2.59 (2.26–3.09) | 1.42 (1.26–1.69) | 1.78 (1.57–2.12) |
| Richer | 1.05 (1.00–1.22) | 1.19 (1.11–1.36) | 1.41 (1.22–1.66) | 1.65 (1.43–1.93) | 1.05 (1.00–1.22) | 1.21 (1.11–1.40) |
| Richest | 1.02 (1.00–1.11)* | 1.01 (1.00–1.09)* | 1.00 (1.00–1.00) | 1.00 (1.00–1.00) | 1.02 (1.00–1.11)* | 1.01 (1.00–1.06)* |
| **Region** | | | | | | |
| Addis Ababa | 1.00 (1.00–1.00) | 1.00 (1.00–1.00) | 1.00 (1.00–1.00) | 1.00 (1.00–1.00) | 1.00 (1.00–1.00) | 1.00 (1.00–1.00) |
| Afar | 4.50 (2.67–7.60) | 5.49 (3.26–8.82) | 16.06 (9.50–27.51) | 20.75 (12.30–33.15) | 4.50 (2.67–7.60) | 5.82 (3.42–9.58) |
| Amhara | 3.11 (1.95–5.16) | 3.69 (2.34–5.58) | 7.07 (4.45–11.61) | 9.41 (5.92–15.02) | 3.11 (1.95–5.16) | 4.13 (2.55–6.69) |
| Benishangul-Gumuz | 3.30 (1.92–5.69) | 4.06 (2.34–6.52) | 6.30 (3.69–10.77) | 8.11 (4.72–13.16) | 3.30 (1.92–5.69) | 4.22 (2.43–7.22) |
| Dire Dawa | 2.60 (1.62–4.34) | 2.91 (1.80–4.57) | 3.26 (2.01–5.43) | 3.69 (2.29–5.85) | 2.60 (1.62–4.34) | 2.94 (1.82–4.77) |
| Gambela | 2.31 (1.59–3.54) | 2.66 (1.80–3.90) | 5.22 (3.58–7.88) | 6.31 (4.28–9.28) | 2.31 (1.59–3.54) | 2.81 (1.92–4.22) |
| Harari | 2.73 (1.86–4.26) | 3.04 (2.08–4.47) | 7.12 (4.82–11.01) | 8.23 (5.60–12.32) | 2.73 (1.86–4.26) | 3.14 (2.12–4.76) |
| Oromia | 2.87 (1.88–4.56) | 3.42 (2.27–5.17) | 8.70 (5.76–13.56) | 11.65 (7.73–17.93) | 2.87 (1.88–4.56) | 3.83 (2.50–6.03) |
| SNNPR | 2.49 (1.64–4.01) | 2.96 (1.97–4.58) | 6.08 (4.05–9.68) | 7.78 (5.17–12.32) | 2.49 (1.64–4.01) | 3.17 (2.06–5.02) |
| Somali | 3.80 (2.39–6.15) | 4.67 (2.92–7.12) | 10.77 (6.74–17.32) | 14.66 (9.18–22.45) | 3.80 (2.39–6.15) | 5.19 (3.22–8.12) |
| Tigray | 2.19 (1.46–3.43) | 2.62 (1.73–3.92) | 3.36 (2.19–5.23) | 4.35 (2.88–6.56) | 2.19 (1.46–3.43) | 2.85 (1.91–4.36) |

**(C) Pneumonia (pneumococcal and Hib)**

*(Continued)*

**Table 3.** (continued)

| | "Counterfactual" | | Baseline | | Maximized vaccination coverage | |
|---|---|---|---|---|---|---|
| | Incidence | Mortality | Incidence | Mortality | Incidence | Mortality |
| **Quintile** | | | | | | |
| Poorest | 2.07 (1.97–2.16) | 2.46 (2.31–2.65) | 2.44 (2.30–2.59) | 3.05 (2.84–3.32) | 2.07 (1.97–2.16) | 2.58 (2.42–2.80) |
| Poorer | 1.76 (1.70–1.81) | 2.13 (2.04–2.27) | 1.96 (1.89–2.04) | 2.48 (2.35–2.67) | 1.76 (1.70–1.81) | 2.23 (2.12–2.38) |
| Middle | 1.50 (1.46–1.55) | 1.71 (1.65–1.79) | 1.69 (1.63–1.75) | 2.00 (1.91–2.12) | 1.50 (1.46–1.55) | 1.78 (1.71–1.88) |
| Richer | 1.05 (1.03–1.07) | 1.16 (1.12–1.21) | 1.12 (1.09–1.15) | 1.23 (1.19–1.29) | 1.05 (1.03–1.07) | 1.16 (1.12–1.21) |
| Richest | 1.00 (1.00–1.00) | 1.00 (1.00–1.00) | 1.00 (1.00–1.00) | 1.00 (1.00–1.00) | 1.00 (1.00–1.00) | 1.00 (1.00–1.01)* |
| **Region** | | | | | | |
| Addis Ababa | 1.00 (1.00–1.00) | 1.00 (1.00–1.00) | 1.00 (1.00–1.00) | 1.00 (1.00–1.00) | 1.00 (1.00–1.00) | 1.00 (1.00–1.00) |
| Afar | 5.99 (5.12–6.96) | 6.21 (5.39–7.35) | 8.36 (7.05–9.88) | 9.27 (7.87–11.07) | 5.99 (5.12–6.96) | 6.64 (5.72–7.87) |
| Amhara | 4.25 (3.82–4.75) | 4.28 (3.83–4.94) | 4.92 (4.40–5.55) | 5.64 (4.91–6.63) | 4.25 (3.82–4.75) | 4.87 (4.26–5.70) |
| Benishangul-Gumuz | 4.94 (4.11–5.88) | 5.11 (4.33–6.30) | 5.43 (4.53–6.49) | 6.10 (5.10–7.57) | 4.94 (4.11–5.88) | 5.55 (4.66–6.87) |
| Dire Dawa | 3.71 (3.20–4.31) | 3.52 (3.07–4.13) | 3.96 (3.40–4.60) | 3.95 (3.41–4.67) | 3.71 (3.20–4.31) | 3.70 (3.22–4.36) |
| Gambela | 2.82 (2.47–3.22) | 2.90 (2.58–3.27) | 3.44 (3.00–3.98) | 3.81 (3.34–4.36) | 2.82 (2.47–3.22) | 3.13 (2.77–3.55) |
| Harari | 3.24 (2.94–3.58) | 3.13 (2.85–3.52) | 3.81 (3.43–4.23) | 3.68 (3.33–4.15) | 3.24 (2.94–3.58) | 3.13 (2.85–3.52) |
| Oromia | 3.56 (3.18–3.99) | 3.67 (3.30–4.18) | 4.55 (4.02–5.16) | 5.44 (4.72–6.35) | 3.56 (3.18–3.99) | 4.24 (3.71–4.90) |
| SNNPR | 3.49 (3.12–3.91) | 3.51 (3.14–4.09) | 4.19 (3.71–4.73) | 4.55 (3.99–5.37) | 3.49 (3.12–3.91) | 3.79 (3.35–4.45) |
| Somali | 4.55 (3.96–5.23) | 4.91 (4.35–5.65) | 5.92 (5.10–6.89) | 7.18 (6.19–8.40) | 4.55 (3.96–5.23) | 5.52 (4.80–6.38) |
| Tigray | 2.99 (2.64–3.38) | 3.11 (2.76–3.50) | 3.20 (2.82–3.62) | 3.74 (3.26–4.26) | 2.99 (2.64–3.38) | 3.50 (3.06–3.99) |

**(D) HPV**

| | "Counterfactual" | Baseline | Maximized vaccination coverage |
|---|---|---|---|
| | Incidence | Incidence | Incidence |
| **Quintile** | | | |
| Poorest | 1.13 (1.09–1.17) | 3.03 (2.93–3.14) | 1.13 (1.09–1.17) |
| Poorer | 1.15 (1.11–1.20) | 2.41 (2.31–2.51) | 1.15 (1.11–1.20) |
| Middle | 1.15 (1.11–1.20) | 2.36 (2.27–2.46) | 1.15 (1.11–1.20) |
| Richer | 1.09 (1.06–1.11) | 1.69 (1.65–1.73) | 1.09 (1.06–1.11) |
| Richest | 1.00 (1.00–1.00) | 1.00 (1.00–1.00) | 1.00 (1.00–1.00) |
| **Region** | | | |
| Addis Ababa | 1.00 (1.00–1.00) | 1.00 (1.00–1.00) | 1.00 (1.00–1.00) |
| Afar | 1.22 (1.14–1.32) | 22.64 (21.01–24.57) | 1.22 (1.14–1.32) |
| Amhara | 1.48 (1.33–1.68) | 12.47 (11.14–14.12) | 1.48 (1.33–1.68) |
| Benishangul-Gumuz | 1.27 (1.17–1.39) | 7.03 (6.47–7.70) | 1.27 (1.17–1.39) |
| Dire Dawa | 1.14 (1.07–1.21) | 3.99 (3.76–4.25) | 1.14 (1.07–1.21) |
| Gambela | 1.31 (1.21–1.43) | 13.73 (12.70–14.98) | 1.31 (1.21–1.43) |
| Harari | 1.24 (1.14–1.36) | 11.88 (10.91–13.03) | 1.24 (1.14–1.36) |
| Oromia | 1.21 (1.12–1.31) | 16.85 (15.60–18.34) | 1.21 (1.12–1.31) |
| SNNPR | 1.12 (1.05–1.19) | 10.66 (9.97–11.37) | 1.12 (1.05–1.19) |
| Somali | 1.06 (1.00–1.11) | 15.68 (14.76–16.46) | 1.06 (1.00–1.11) |
| Tigray | 1.35 (1.23–1.51) | 5.86 (5.28–6.54) | 1.35 (1.23–1.51) |

* This population sub-group did not have the lowest risk in all simulations, which is reflected by a slightly wider 95% uncertainty range.

relative distributions of morbidity and mortality, by developing a linear system of equations for estimation of the joint distribution of risk and prognostic factors. By treating each possible combination of risk or prognostic factors as computationally distinct, we are able to account for individuals with multiple risk factors. While more realistic, the resulting relative risk

gradients are in the same general range as those found by Chang and colleagues [16], but the gradients are slightly greater. This makes sense at the individual level, since having multiple risk factors increases one's risk of morbidity and mortality compared to having a single risk factor, and at the population subgroup level, since the poor and those living in more rural regions are more likely to have multiple risk factors and often lack access to basic health care services.

However, our modeling approach is limited by the way in which relative risks are measured. The standard methodology compares an individual's risk of morbidity or mortality with a given factor to a null scenario (i.e., no risk), whereas a more realistic evaluation would compare risk factors in various scenarios (e.g., comparing the risk related to having multiple factors with the risk related to having a single factor or a different set of factors). This model develops a more realistic depiction of how risk and prognostic factors can combine to produce morbidity and mortality without completely reworking the standard way in which relative risks are calculated. Additionally, this model does not include dynamic transmission of the infectious diseases examined within and across socioeconomic groups [35], and it therefore does not consider some meaningful factors that can contribute to disease incidence, such as household crowding, population density, and social contact among population subgroups. Lastly, quality of care can play an important role in mortality outcomes in addition to treatment-seeking behavior; however, data were not sufficiently available to include such a quality factor in our modeling.

Our results show that current unequal levels and distributions of vaccination and treatment-seeking patterns do not necessarily redress health inequalities across wealth quintiles and geographic regions. The poor and those in more rural regions are at a greater risk of incidence and mortality under the "counterfactual" scenario, because of the higher prevalence of risk and prognostic factors and lower health-seeking behavior among these population subgroups. In *absolute* terms, all populations subgroups will benefit from health interventions–in particular, the poorest will likely benefit the most in terms of absolute numbers of disease cases and deaths averted by vaccines. However, the relative distribution of health interventions across population subgroups may increase existing *relative* disparities (i.e., among the remaining cases and deaths not averted by vaccines). This is clearly not a reflection of the health interventions themselves but is rather due to unequal access to care (often pro-rich gradients) through vaccination coverage and treatment-seeking behavior.

This can be observed in the case of HPV: the poorest quintile is at an estimated 3.03 times greater risk of disease than the richest quintile in the baseline scenario (with vaccination and treatment-seeking behavior), compared to 1.13 times greater risk in the "counterfactual" scenario (no health interventions). For deaths, the greatest difference occurs for measles: the poorest quintile is at an estimated 4.20 times greater risk of death than the richest in the baseline scenario compared to 2.34 times greater risk in the "counterfactual" scenario. This is likely most apparent for these two VPDs, because of the corresponding vaccines' high efficacy (85% for MCV and 100% for HPV vaccine).

These findings highlight the urgent need to target both (1) the underlying risk factors (and corresponding social determinants) that contribute to VPD incidence and set the foundation for health inequalities in Ethiopia, and (2) low vaccination coverage and low treatment-seeking behavior, as lower rates among the poor and those in rural regions exacerbate relative inequalities. This analysis highlights that poor and rural children in Ethiopia stand to benefit the most from more equitable vaccination programs because they would have the most to gain. Even pro-rich vaccination coverage, in which the richest quintiles have higher coverage rates (as observed in Ethiopia), would likely result in greater health benefits in absolute terms (i.e., number of infections and deaths averted) given the more severe risk and prognostic profiles of

the poor [9]. Ethiopia's vaccination program could become more pro-poor if coverage were more equal across wealth quintiles and the rural-urban divide, but might maintain relative health inequities if current pro-rich vaccine coverage rates remain.

The WHO's "Reach Every District" (RED) approach outlines key operational components specifically aimed at improving immunization coverage in every district [36]. Stronger implementation of this approach, which includes routine supervision of health facilities and monitoring for action, would likely improve disparities in Ethiopia's vaccination coverage. Indeed, in Ethiopia, DPT3 coverage has been found to be more than 4 times greater in zones (the administrative level below region) that conduct health facility supervision, provide written feedback to their health facilities, and have health facilities that monitor their immunization performance [37]. Beyond the health system, evidence in Ethiopia suggests that improving women's autonomy through education and gender-related initiatives can also lead to improved immunization coverage [38].

Our study presents a number of limitations. First is the somewhat outdated (from the DHS year 2016) and lacking evidence on risk and prognostic factors, especially for HPV, and the way in which these factors are estimated (i.e., compared to a null scenario, discussed above). Much has happened since the 2016 Ethiopian DHS (e.g., the COVID-19 pandemic, armed conflict); however, the 2016 Ethiopian DHS remains the most recent source of comprehensive disaggregated (by wealth group and geographic region) empirical data on risk and prognostic factors relevant to this analysis. Additionally, the epidemiological link between VPD cases and deaths is likely much more complicated than any of the three methods used by our modeling to estimate relative mortality risk gradients. The second relates to the model's optimization. The optimization was performed from a random starting point, and therefore there is no guarantee of stable results; and because the minimizer is not unique, there is no assurance that the scheme samples uniformly amongst the minimizers. Uniform sampling from high dimensional convex spaces can be computationally challenging, so we adopted a less sophisticated rearrangement method based on independent samples for each variable to generate starting points. Furthermore, the mathematical methods operate under the assumption that the proportional distribution across quintiles and regions are invariant under a scaling of the population from the survey sample. Rounding cutoffs and small perturbations of survey population were employed for computational reasons. Lastly and importantly, we lacked the empirical data (e.g., relative distributions in the incidence and mortality of those VPDs across socioeconomic groups in Ethiopia) to validate the model.

Nevertheless, the methodology employed by this research takes simultaneously into account the joint nature of VPD risk and prognostic factors and uses the distribution of these factors to estimate relative gradients of VPD morbidity and mortality across population subgroups, wealth quintiles and geographic regions, in Ethiopia. While vaccination undoubtedly improves health outcomes in absolute terms, especially so among the poorest, the results of our modeling highlight the fact that unequal access to healthcare interventions can exacerbate pre-existing inequalities in the relative distribution of the risk and prognostic factors that give rise to VPD burden (among diseases cases and deaths not averted by vaccines). Current unequal levels and pro-rich gradients of vaccination and treatment-seeking patterns should be redressed so to significantly improve health equity across wealth quintiles and geographic regions in Ethiopia.

## Supporting information

**S1 Text.**
(DOCX)

## Acknowledgments

We thank Iman Ali for research assistance. We are grateful to the Maternal, Child Health & Nutrition Directorate, Ministry of Health, Ethiopia and participants to a vaccine economics webinar (January 2021) for their engagement and valuable comments including Dr. Meseret Zelalem Tadesse (Director) and Mr. Yohannes Lakew Tefera (Deputy Director).

## Author Contributions

**Conceptualization:** Stéphane Verguet.

**Data curation:** Sarah Bolongaita, Dominick Villano.

**Formal analysis:** Sarah Bolongaita, Dominick Villano, Stéphane Verguet.

**Investigation:** Sarah Bolongaita, Dominick Villano, Solomon Tessema Memirie, Sophia Comas, Eva Rumpler, Stephanie M. Wu, Ryoko Sato.

**Methodology:** Dominick Villano, Stéphane Verguet.

**Software:** Sarah Bolongaita, Dominick Villano.

**Supervision:** Stéphane Verguet.

**Validation:** Sarah Bolongaita, Dominick Villano, Solomon Tessema Memirie, Mizan Kiros Mirutse, Alemnesh H. Mirkuzie, Angela Y. Chang, Stéphane Verguet.

**Visualization:** Sarah Bolongaita.

**Writing – original draft:** Sarah Bolongaita, Solomon Tessema Memirie.

**Writing – review & editing:** Sarah Bolongaita, Dominick Villano, Solomon Tessema Memirie, Mizan Kiros Mirutse, Alemnesh H. Mirkuzie, Sophia Comas, Eva Rumpler, Stephanie M. Wu, Ryoko Sato, Angela Y. Chang, Stéphane Verguet.

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
