## [Decision Letter · Decision Letter 0]

2 Mar 2022

PGPH-D-22-00010

Estimating the risk of incidence and mortality of select vaccine-preventable diseases by wealth group and geographic region in Ethiopia

Dear Dr. Bolongaita,

Thank you for submitting your manuscript to PLOS Global Public Health. After careful consideration, we feel that it has merit but does not fully meet PLOS Global Public Health’s publication criteria as it currently stands. Therefore, we invite you to submit a revised version of the manuscript that addresses the points raised during the review process.

In particular, please explain any differences in your paper's methodology and that of Chang 2018 (see reviewer 1).

In your response to reviewer, please specify, in response to every comment, what specific edit you made (paste in the sentence(s)), and where it is in the revised manuscript.

We look forward to receiving your revised manuscript.

Kind regards,

Abram Luther Wagner, PhD, MPH

Academic Editor

Journal Requirements:

1. We ask that a manuscript source file is provided at Revision. Please upload your manuscript file as a .doc, .docx, .rtf or .tex. If you are providing a .tex file, please upload it under the item type ‘LaTeX Source File’ and leave your .pdf version as the item type ‘Manuscript’.

2. Please provide separate figure files in .tif or .eps format only and remove any figures embedded in your manuscript file. Please ensure that all files are under our size limit of 20MB. If you are using LaTeX, you do not need to remove embedded figures.

3. We notice that your supplementary figures and tables are included in the manuscript file. Please remove them and upload them with the file type 'Supporting Information'. Please ensure that all Supporting Information files are included correctly and that each one has a legend listed in the manuscript after the references list. 

4. Please provide us with a direct link to the base layer of the map used in Figures 2 and 3, and ensure this location is also included in the figure legend. 

Please note that, because all PLOS articles are published under a CC BY license (creativecommons.org/licenses/by/4.0/), we cannot publish proprietary maps such as Google Maps, Mapquest or other copyrighted maps. If your map was obtained from a copyrighted source please amend the figure so that the base map used is from an openly available source.

Please note that only the following CC BY licences are compatible with PLOS licence: CC BY 4.0, CC BY 2.0  and CC BY 3.0, meanwhile such licences as CC BY-ND 3.0 and others are not compatible due to additional restrictions. If you are unsure whether you can use a map or not, please do reach out and we will be able to help you. 

The following websites are good examples of where you can source open access or public domain maps:

Additional Editor Comments (if provided):

Please respond to the reviewer comments. In particular, please explain any differences between your papers' methodology and the Chang 2018 paper.

Reviewers' comments:

Reviewer's Responses to Questions

**Comments to the Author**

1. Does this manuscript meet PLOS Global Public Health’s publication criteria? Is the manuscript technically sound, and do the data support the conclusions? The manuscript must describe methodologically and ethically rigorous research with conclusions that are appropriately drawn based on the data presented.

Reviewer #1: Yes

Reviewer #2: Partly

2. Has the statistical analysis been performed appropriately and rigorously?

Reviewer #1: Yes

Reviewer #2: No

3. Have the authors made all data underlying the findings in their manuscript fully available (please refer to the Data Availability Statement at the start of the manuscript PDF file)?

Reviewer #1: Yes

Reviewer #2: Yes

4. Is the manuscript presented in an intelligible fashion and written in standard English?

Reviewer #1: Yes

Reviewer #2: Yes

5. Review Comments to the Author

Reviewer #1: In this manuscript, the authors employ an equity perspective and estimate the risk of incidence and mortality in selected vaccine-preventable diseases by household wealth groups and geographic regions in Ethiopia. In this modeling exercise, data from the Ethiopian Demographic and Health Survey were used to generate a simulated population, and the methodology was based on earlier published work. The authors present the modeling methodology and discuss its limitations.

The empirical data used in the modeling emanate from the Ethiopian Demographic and Health Survey 2016, including data from almost a decade ago. The authors do not discuss the consequences for their estimates by the development during the years that have passed; first, increased coverage, followed by social insecurity, war, and pandemic.

Inequities in coverage of health interventions depend on context, e.g., population and health system characteristics. This information is lacking in the manuscript. A minimum would be to briefly describe the health system and how vaccination services are organized in rural vs. urban areas and pastoralist vs. agrarian regions.

The authors conclude that their modeling shows that, given the inequities in using services, the health interventions (vaccination, treatment) worsen disparities. This result is not new information. It has been discussed in equity-focused publications from Ethiopia and Government policy documents for a relatively long time. My question is whether the authors may extract and discuss more policy-relevant conclusions?

Reviewer #2: 1. The model simulates three scenarios – 1) no vaccination/ treatment seeking (counterfactual), 2) current vaccination and treatment seeking (baseline), and 3) maximized vaccination. It is counter intuitive why health outcomes are worsened in the baseline scenario. This was not the case for results reported by Chang et al. (2018). Since the current study was built on Chang et al.’s framework, please provide justification of this counter intuitive results.

2. It is stated that “health interventions worsen the underlying inequalities observed in the counterfactual scenario”. What is the mechanism of this result?

3. Does it mean abandoning vaccination programs will improve health outcomes? For example, Table S9 reports that at the poorest quintile, death from Rotavirus diarrhea is 1.64 in counterfactual and 2.35 in maximized coverage. Please explain this result. Also discuss what features of the model are driving this counter intuitive estimates.

4. Could you offer any theoretical explanation of the counter intuitive findings?

5. Could you report the aggregate incidence and mortality for each VPD in Table 3, in addition to results by quintiles and regions.

6. What is the underlying mechanism of the model? Equitable vaccination coverage or higher vaccination coverage as a whole?

7. Could you check another scenario with coverage rate be at the minimum (i.e., of the lowest quintile)? It is unclear from the analysis that whether more coverage is better or equal coverage is better. I think comparison of this new scenario with the baseline and maximized scenarios may provide some new perspectives.

6. PLOS authors have the option to publish the peer review history of their article (what does this mean?). If published, this will include your full peer review and any attached files.

**Do you want your identity to be public for this peer review?** For information about this choice, including consent withdrawal, please see our Privacy Policy.

Reviewer #1: **Yes: **Lars Ake Persson

Reviewer #2: No

---

## [Editor Report · Decision Letter 1]

18 Jul 2022

Estimating the relative risk of incidence and mortality of select vaccine-preventable diseases by wealth group and geographic region in Ethiopia

PGPH-D-22-00010R1

Dear Ms Bolongaita,

We are pleased to inform you that your manuscript 'Estimating the relative risk of incidence and mortality of select vaccine-preventable diseases by wealth group and geographic region in Ethiopia' has been provisionally accepted for publication in PLOS Global Public Health.

Best regards,

Abram L. Wagner, PhD, MPH

Academic Editor
